# Epigenetic Distribution of Recombinant Plant Chromosome Fragments in a Human–*Arabidopsis* Hybrid Cell Line

**DOI:** 10.3390/ijms22115426

**Published:** 2021-05-21

**Authors:** YengMun Liaw, Yikun Liu, CheeHow Teo, Petr Cápal, Naoki Wada, Kiichi Fukui, Jaroslav Doležel, Nobuko Ohmido

**Affiliations:** 1Graduate School of Human Development and Environment, Kobe University, Kobe, Hyogo 657-8501, Japan; liawlym@gmail.com (Y.L.); liuyikun9410@yahoo.co.jp (Y.L.); 2Centre for Research in Biotechnology for Agriculture, Universiti Malaya, Lembah Pantai, Kuala Lumpur 50603, Malaysia; cheehow.teo@um.edu.my; 3Institute of Experimental Botany of the Czech Academy of Sciences, Centre of the Region Hana for Biotechnological and Agricultural Research, Šlechtitelů 31, 779 00 Olomouc, Czech Republic; capal@ueb.cas.cz (P.C.); dolezel@ueb.cas.cz (J.D.); 4Graduate School of Technology, Industrial and Social Sciences, Tokushima University, Kuramoto, Tokushima 770-8503, Japan; wada.naoki@tokushima-u.ac.jp; 5Graduate School of Pharmaceutical Sciences, Osaka University, Suita, Osaka 565-0871, Japan; kfukui@bio.eng.osaka-u.ac.jp

**Keywords:** *Arabidopsis* genome, DNA methylation, epigenome, human–plant hybrid cell line, gene expression, whole-genome bisulfite sequencing (WGBS)

## Abstract

Methylation systems have been conserved during the divergence of plants and animals, although they are regulated by different pathways and enzymes. However, studies on the interactions of the epigenomes among evolutionarily distant organisms are lacking. To address this, we studied the epigenetic modification and gene expression of plant chromosome fragments (~30 Mb) in a human–*Arabidopsis* hybrid cell line. The whole-genome bisulfite sequencing results demonstrated that recombinant *Arabidopsis* DNA could retain its plant CG methylation levels even without functional plant methyltransferases, indicating that plant DNA methylation states can be maintained even in a different genomic background. The differential methylation analysis showed that the *Arabidopsis* DNA was undermethylated in the centromeric region and repetitive elements. Several *Arabidopsis* genes were still expressed, whereas the expression patterns were not related to the gene function. We concluded that the plant DNA did not maintain the original plant epigenomic landscapes and was under the control of the human genome. This study showed how two diverging genomes can coexist and provided insights into epigenetic modifications and their impact on the regulation of gene expressions between plant and animal genomes.

## 1. Introduction

Epigenetic modifications play an important role in genome functions and their responses to environmental stimuli. The modifications are heritable and reversible and ultimately affect a broad range of processes, extending from the regulation of individual genes to the spatial genome organization in cell nuclei [1]. They also have an important role in controlling the activity of transposable elements (TEs) and genome integrity [2]. The targets of epigenetic modifications include DNA, which can be methylated, histones, which can be modified in many different ways, and noncoding RNA [1,2]. The methylation of DNA involves the addition of a methyl group (-CH_3_) to the cytosine base, usually at the fifth position, forming 5′-methylcytosine. In the mammalian genome, DNA methylation occurs in almost two-thirds of CG dinucleotides, whereas nonmethylated CG pairs are less abundant [3,4]. In plants, DNA methylation occurs within the CG, CHG, and CHH sequence contexts (H = A, C, or T), mainly in repetitive DNA [5]. DNA methylation can control gene expression in a tissue-specific manner, thus ensuring the proper growth, development, and function of an organism. Since the 1980s, researchers have associated the demethylation of genes and DNA repeats with uncontrolled growth and cancer in humans [6]. For example, the genomic methylation levels in colon cancer cells are significantly lower (8−10%) than those in normal tissue [7]. Besides cancer, DNA methylation was shown to be associated with other human diseases, such as autoimmune diseases, neurological disorders, and metabolic disorders [8]. Epigenomic profiling thus enables a better understanding of epigenetic control on a genome-wide scale.

In plants and animals, DNA methylation is regulated by different pathways and enzymes. In mammals, the methylation landscape is controlled by members of the DNA methyltransferase (Dnmt) family, in which de novo methylation is catalyzed by Dnmt3 and maintained by Dnmt1 [9]. In *Arabidopsis*, de novo methylation is established by DRM2 (domain-rearranged methyltransferase 2) via the RNA-directed DNA methylation (RdDM) pathway. Methylation in the context of CG, CHG, and CHH is maintained by MET1 (methyltransferase 1), CMT3 (chromomethylase 3), and DRM2, respectively [4,10], with DRM2 and CMT3 functioning redundantly at the CHH sites. Although controlled by different pathways, methylation patterns in plants and animals exhibit a certain level of conservation [11]. *MET1* is the plant homolog of *Dnmt1*, whereas *DRM2* is the plant homolog of *Dnmt3* [4,12]. The CMT class of methyltransferases is unique to the plant kingdom [13,14]. Although humans and plants diverged 1.5 billion years ago, their methylation systems are similar, as indicated by the homology between their methyltransferases [15]. However, the absence of a proper methylation system is more lethal to mammals than to plants [4].

Methylation changes can occur in an organism in response to environmental changes and stress [16], as well as during hybridization in plants [17] and speciation in animals [18]. Previous studies on epigenetic changes were performed on closely related species and their hybrids or on a small scale involving insertions of up to kilobases in size [18,19,20]. Particularly, the insertion of foreign DNA can cause methylation changes in both the donor and the recipient genomes [19,21], leading to alterations in the methylation profile of the recipient genome [22]. Weber et al. [23] showed genome-wide methylation changes after the insertion of a 5.6-kb plasmid into the human genome, indicating the destabilization of the epigenome due to the insertion of foreign DNA. Regarding the insertion of large foreign DNA fragments into diverged species, a recent study on mouse and human cell lines carrying several megabases of yeast DNA showed different chromatin condensation patterns compared to those in the surrounding host mammalian DNA [24]. This indicated that the maintenance of the foreign chromatin was different compared to the host cell. The interactions among and functional conservation of epigenomes among evolutionarily distant organisms have also not been widely studied to date.

In our previous work, Wada et al. [25] created a human–plant hybrid cell line by fusing *Arabidopsis thaliana* protoplasts and human HT1080 cells. A neo-chromosome was formed by the insertion of segments of Arabidopsis chromosomes into human chromosome 15. The human–Arabidopsis hybrid cells stably maintained the plant-derived neo-chromosomes (PD chromosomes), and a number of Arabidopsis genes were expressed in the human genetic background. An analysis of the neo-chromosome in 60- and 300-day-old cell cultures by next-generation sequencing and molecular cytogenetics suggested its origin by the fusion of DNA fragments of different sizes from Arabidopsis chromosomes 2–5, which were randomly intermingled rather than joined end-to-end [26].

To address this knowledge gap, we performed whole-genome bisulfite sequencing (WGBS) of a 300-day-old human–*Arabidopsis* hybrid cell line and gene expression analysis on 60-day-old and 300-day-old hybrid cell lines. In this cell line, large *Arabidopsis* genome fragments (~30 Mb) were maintained in the human cell background during a long-term in vitro culture [25,26]. This analysis allowed us to characterize the epigenetic status and gene expression of the plant chromosome fragments integrated into the human genome. The study of the epigenetic regulation of foreign DNA in this hybrid cell line would be useful to understand the coexistence of two divergent genomes and provide insight into the fundamental principles underlying genome interactions beyond their biological kingdoms. This research could shed light on certain fundamental principles of epigenetic changes and modifications that might be useful in some heterologous systems for biological applications.

## 2. Results

### 2.1. Genome-Wide Methylation Landscape of Plant Genome Fragments in Human Hybrid Cells

To examine the epigenetic landscape of the *Arabidopsis* chromosome fragments integrated into a human chromosome, WGBS was performed on a 300-day-old *Arabidopsis*–human hybrid cell line in vitro. This hybrid line harbored large genome segments originating from *Arabidopsis* chromosomes 2, 3, and 5 translocated to human chromosome 15 [26]. The *Arabidopsis* and human genome regions in the hybrid cell line were compared to the *Arabidopsis* reference genome (TAIR10) and human genome (GRCh38), respectively (Figure 1). After trimming, 1,082,071,072 reads with 20.36% GC were retained, with 96.98% in Q20 and 91.45% in Q30. The bisulfite conversion rate was estimated to be 99.75%, using lambda phage DNA as the spike-in control. In total, 0.35% (3,751,598) of the uniquely mapped reads were obtained from TAIR10 mapping. Qualimap 2.2 showed a mean coverage of 3.2× across TAIR10. After deduplication, 2,847,298 reads were processed for methylation calling in BSMAP [27] and Bismark [28]. The genome-wide methylation profiles for the *Arabidopsis* regions were 23.8%, 0.27%, and 0.26% in the context of CG, CHG, and CHH, respectively (Figure 1). Similar methylation profiles were obtained using Bismark, with 23.8%, 0.3%, and 0.3% in the context of CG, CHG, and CHH, respectively (Appendix A). The mapping efficiency of Bismark was 0.3%, with 1,780,771 paired-end reads of the unique best hits. The low mapping efficiency was due to the relatively small fraction of the *Arabidopsis* genome (~30 Mb) in the human genome background (~3 Gb) [26].

The extent of methylation of CG in the introgressed *Arabidopsis* genome regions was similar to that of wild-type *Arabidopsis* leaf tissue, retaining 24% CG methylation [29]. However, the frequency of CHG and CHH in the recombinant *Arabidopsis* genome region was 0.3%, similar to that of human DNA (Figure 1). Therefore, CG methylation remained at the same levels as those in *Arabidopsis* plant tissues, whereas the frequency of CHG and CHH decreased to the level observed in humans and, hence, followed the host methylation patterns (Figure 1).

To determine whether the human epigenome landscape was disrupted by the insertion of large, foreign DNA, the WGBS data were aligned to the human genome reference (GRCh38) to determine the global methylation level in the human genome. For the human genome region in the hybrid cell line, the methylation status of CG, CHG, and CHH was similar to that of the original HT1080 cell line (CG, 67%, CHG, 0.2%, and CHH 0.4%) [30], indicating that the insertion of a large, alien DNA fragment (~30 Mb) did not affect the DNA methylation of the host genome (Figure 1).

### 2.2. Patterns of DNA Methylation in Genes and TEs

Next, we examined the methylation levels across the gene regions 2-kb upstream and downstream of the transcription start and end sites and TEs in the hybrid cells. The genes and TEs displayed different methylation patterns (Figure 2). In the gene regions, CG methylation was lowest near the transcription start site (TSS) and was gradually increased across the gene body with a sharp decrease at the transcription end site (TES). However, the TEs showed similar methylation levels across their gene bodies, with 2-kb upstream and downstream sequences. CHG and CHH methylation across the genes and TEs were both close to zero (Figure 2).

### 2.3. Differentially Methylated Arabidopsis Genome Segments in Hybrid Cell Lines

To assess the similarities in *Arabidopsis* DNA methylation between the wild-type leaf tissue and the hybrid line, two wild-type *Arabidopsis* leaf tissue WGBS datasets (SRR7596644 and SRR534177) were downloaded as the controls from SRA (https://www.ncbi.nlm.nih.gov/sra, accessed on 19 May 2021), and Pearson pairwise correlations were calculated using methylKit [31]. The correlation coefficients between both wild-type controls were high at 0.97 but low with respect to the hybrid cell line (0.21 and 0.19), indicating that the methylation of the *Arabidopsis* regions in the hybrid cells was very different from that of wild-type *Arabidopsis* (Figure 3a). SRR7586644 was used as the wild-type control for the following experiments based on its paired-end sequencing format, which is more similar to the hybrid cell data and has a slightly better sequence quality.

To determine the differentially methylated *Arabidopsis* chromosome regions in the hybrid cells, the extent of DNA methylation within a 1-kb window was calculated, and the differentially methylated regions (DMRs) of the introgressed *Arabidopsis* DNA were identified with the criteria of a methylation difference of at least 25% in the Fisher’s exact test and a *p*-value < 0.01. The *Arabidopsis* DNA in the hybrid cells harbored 6% more hypomethylated regions than hypermethylated regions. We further investigated the distribution of differentially methylated regions on the fragments originating from different *Arabidopsis* chromosomes and their locations in these fragments. Overall, the hypermethylated genomic regions were distributed similarly across the three main *Arabidopsis* chromosomes that were present in the 300-day-old hybrid cell line, with chromosomes 2, 3, and 5 containing ~15% hypermethylated regions (Figure 3b). The number of hypomethylated regions on chromosome 5 were twice that of the hypermethylated regions on the same chromosome. The levels of hypermethylated and hypomethylated regions were similar for chromosomes 2 and 3. Differentially methylated regions on the remaining chromosomes 1 and 4 were not considered because of the low number of bases aligned to the *Arabidopsis* genome. Plotting differentially methylated regions with respect to the chromosomal location showed a negative value for the methylation differences around the centromeric and pericentromeric regions of chromosome 5 (Figure 3c), indicating the presence of a hypomethylated *Arabidopsis* centromere.

The differentially methylated regions were annotated based on the gene annotation data from Araport (The Arabidopsis Information Portal (https://www.araport.org, accessed on 19 May 2021)) to determine the percentage of differentially methylated regions located in the promoter/intron/exon/intergenic regions. In total, 61% of the differentially methylated regions were located in the promoter regions, with the rest covering the intergenic regions (20%) and exons (20%). A gene ontology analysis did not show any preferential enrichment in the gene families.

### 2.4. Hypomethylation of Arabidopsis Repetitive Elements in the Human Genome Background

To determine whether the *Arabidopsis* DNA repeats were preferentially methylated in the human genome background, the differentially methylated regions were examined against the overlapping regions based on the RepeatMasker tracks in the ReMap database [32]. Interestingly, all the *Arabidopsis* repeat families (LINE, LTR, DNA transposons, satellites) showed hypomethylation in the hybrid cell line (Figure 4). The most distinct pattern was observed in the satellite repeat families, with the hypomethylation being 60 times higher than that in the hypermethylated regions. Within the satellite family, the COLAR12 class, which is the consensus sequence of 178 bp of *Arabidopsis* satellites restricted to pericentromeric heterochromatin [33], showed the highest hypomethylation (Appendix A). This was followed by LTR regions with a 35-fold difference. The highest frequency of LTR elements in the hypomethylated regions was in the Athila LTR family (Appendix A), which is a centromere-associated LTR element.

### 2.5. Function of Arabidopsis Methyltransferase in Hybrid Cells

In *Arabidopsis*, cytosine methylation in different contexts is maintained by specific methyltransferases, with CG methylation being maintained by *MET1*, CHG by *CMT3*, and CHH by *DRM2* (Table 1). We investigated the *Arabidopsis* methyltransferase genes maintaining CG, CHG, and CHH methylation though the WGBS read alignment data, polymerase chain reaction (PCR), and reverse transcription-PCR (RT-PCR). *CMT3* was not used for the PCR and RT-PCR analyses because of the absence of read alignments based on the WGBS data. The investigated *Arabidopsis* methyltransferases were either absent or not expressed in the hybrid cell line (Figure 5 and Table 1). *MET1* and *DRM2*, responsible for maintaining CG and CHH methylation, were not expressed (Figure 5 and Table 1), whereas *CMT3*, responsible for maintaining CHG methylation, was absent in the hybrid cell line based on the alignment data.

### 2.6. Expression Analysis of Plant-Specific Genes in Hybrid Cells

To investigate the expression of plant genes in the hybrid cell line, ten *Arabidopsis* genes were selected to test for their expression based on biologically interesting functions, such as photosynthesis and cell wall biosynthesis (Table 2). The results of the RT-PCR analysis partially agreed with the previous microarray data [25], with six (*NPY4, CA2, LHCB4.3*, *AtNADH*, *GAMMA-TIP*, and *PLP6)* of the 10 genes expressed in both the 60- and 300-day-old hybrid cell lines (Table 2 and Figure 6). The presence or absence of gene expression was consistent between the 60- and 300-day-old hybrid cells (Table 2 and Figure 6). The *HPR* gene, located on *Arabidopsis* chromosome 1, was eliminated in the 300-day-old hybrid cells and was not expressed in the 60-day-old hybrid cells. Three highly expressed genes in the microarray analysis (*AtNADH*, *GAMMA-TIP*, and *PLP6*) that were validated previously using RT-PCR in the 60-day-old hybrid cells [25] were expressed in both the 60- and 300-day-old hybrid cells, suggesting that they might remain in the expression during the 240-day culture (Figure 6). The expression of two cell wall-related genes, *CESA4* and *GSL8*, was also investigated in the hybrid cell lines. We found that, although both genes were present, they were not expressed in the 60- and 300-day-old hybrid cells.

## 3. Discussion

We investigated the epigenetic profiles and transcription of the large plant-derived chromosome fragments in the human–plant hybrid cell line established by Wada et al. (2017) [25]. In our previous study, we performed whole-genome sequencing of the human–*Arabidopsis* hybrid cell line in vitro, in which large *Arabidopsis* genome fragments (~30 Mb) were maintained in the human cell background for 300 days [26]. This study, using the same hybrid cell line, revealed changes in epigenetic modifications of the plant chromosome fragments in a human genome background that occurred at CG- and non-CG methylation sites (Figure 1) and the expression status of the plant-specific methyltransferases (Figure 5) and plant-specific genes (Table 2 and Figure 6) under human genome regulation in the hybrid cell line. DNA methylation of the introgressed *Arabidopsis* genome fragments was controlled by the human genome apparatus, with non-CG methylation patterns similar to those of the human host (Figure 1). This is consistent with the fact that *Arabidopsis* methyltransferases were not expressed (Figure 5). The analysis of differential methylation of the *Arabidopsis* chromosome fragments revealed a hypomethylated centromeric region and repetitive elements in hybrid cells compared to those in the wild-type *Arabidopsis* leaf tissue (Figure 3b,c and Figure 4).

### 3.1. Conservation of Genome-Wide Arabidopsis CG Methylation Level in Human Genome Background

The extent of CG methylation of *Arabidopsis* chromosome fragments in the hybrid cells was similar to that in wild-type *Arabidopsis* [29,34,35]. The preservation of CG methylation patterns of the translocated chromosomal fragments was consistent with the observations of previous studies in vertebrates, demonstrating that the original sequence context of the CG sites is sufficient to determine the DNA methylation pattern after replication in a new host genome [36,37]. When hundreds of foreign DNA fragments (5.7 kb in reference [38] and 100–600 bp in reference [39]) were previously inserted into the same genomic locus in mouse cells, most of the inserts re-established their original methylation states. Even a whole human chromosome when inserted into a mouse cell line can maintain almost 80% of its native epigenetic state [36]. Furthermore, Long et al. (2016) [36] observed a similar phenomenon in more distantly related vertebrates; for example, mouse chromosomal fragments inserted into zebrafish embryos retained their original methylation states. Collectively, these studies demonstrated the existence of “DNA methylation grammar”, indicating that sequence contexts alone are sufficient to maintain the original DNA methylation states, irrespective of the genomic environment [37]. Although these studies were performed in vertebrates with a higher density of CG islands than that in plants, considering the similar CG methylation percentage of plant DNA in a human genome background, similar rules of methylation conservation might be applicable to plant DNA.

In contrast, the non-CG methylation patterns of the introgressed *Arabidopsis* chromosome segments were more similar to the human methylation patterns, probably due to the absence of functional *Arabidopsis* methyltransferases in the human background (Figure 5). The suppressed expression of *Arabidopsis* methyltransferases and low non-CG methylation patterns indicated repression by the host human genome. Two mechanisms were proposed to explain the reduction in the DNA methylation levels, active demethylation by demethylase, and passive dilution of methylation during DNA replication [40]. The active demethylation of *Arabidopsis* DNA could play a role, as Gallego-Bartolomé et al. (2018) [41] demonstrated; the catalytic domain of the human demethylase *ten-eleven translocation 1* (*TET1*) causes targeted demethylation in *Arabidopsis* genes. This showed that enzymes from divergent species can function in different environments. However, human *TET1* was not active with respect to non-CG site demethylation [42], and thus, the reduction in non-CG methylation was probably due to the absence of de novo methylation, required to maintain the existing methylation pattern during the division of the hybrid cells. Therefore, we concluded that the passive dilution of non-CG methylation, rather than the regulation of active demethylation, occurred in the hybrid cell line. The RdDM pathway is responsible for the regulation of non-CG methylation in plants [43,44]. In plants, the RdDM pathway is also used to silence transposable elements, whereas this pathway is absent in humans [45]. RdDM depends on the production of 24-nucleotide siRNA to direct repressive epigenetic marks to the target region. The absence of this pathway might explain why the Arabidopsis repetitive elements were hypomethylated in the hybrid.

Methylation across the genes and TEs of *Arabidopsis* DNA in the hybrid cells displayed patterns similar to plant and animal methylation, respectively. CG methylation across the genes was high at 2-kb upstream, decreased at the TSS, and gradually increased across the gene bodies before a sharp decrease at the TES (Figure 2). The methylation pattern across the repetitive DNA, which was similar to that in the mammalian genome [34], might suggest the regulation of repetitive elements in the host human genome to ensure genome integrity.

### 3.2. Hypomethylation of Arabidopsis Centromere in 300-Day-Old Hybrid Cells

One of the key findings of this study was the observation of hypomethylation of the translocated plant centromeric and pericentromeric regions in the hybrid line (Figure 2). A functional centromere in a given species is defined not only by centromeric DNA but, also, by the epigenetic environment. Functional plant centromeres are often characterized by a hypomethylated core centromere sequence and hypermethylated pericentromere [46]. Thus, the methylation pattern of the plant centromere in hybrid cells differs from that of a functional plant centromere. In humans, a histone H3 variant CENP-A is important for centromere identity and function [47]. Wada et al. (2017) [25] detected human CENP-A, a marker of active centromeres, on two subtypes of the plant-derived chromosome (types S and A) without any human centromeric sequences, indicating that functional centromeres might be formed in regions containing plant centromeric DNA in hybrid cells. However, CENP-A was detected only in human centromeres but not in *Arabidopsis* centromere repeats on a dicentric translocation chromosome (type T), which was mainly maintained in the hybrid cells used in this study, suggesting that *Arabidopsis* centromeres were not functional in the type T chromosome analyzed. Dicentric chromosomes are unstable, although a stable pseudodicentric chromosome might be formed when one of the centromeres is inactivated [48]. We concluded that the *Arabidopsis* centromere repeats lose their function because of the methylation regulation by the human genetic apparatus, leaving only one active centromere of human origin on the hybrid chromosome.

### 3.3. Hypomethylation of Arabidopsis Repetitive Elements in the Human Genome Background

Repetitive sequences of *Arabidopsis* origin were also hypomethylated in the hybrid cell line (Figure 4). TEs were found to be hypermethylated during environmental stress in plants and hypomethylated in human cancer cells [3,49]. The hypomethylation of TEs also functions as a post-zygotic isolation mechanism during nascent species speciation [18]. The methylation analysis in hybrid tilapia revealed differential methylation of the sex determination chromosomal region [50]. In an interspecific marsupial hybrid, the genome-wide undermethylation of retrotransposons, as compared to that in the parental species, demonstrated that speciation can occur via a decrease in the methylation of repetitive elements [20]. These examples demonstrated the hypomethylation of repetitive elements occurring during differentiation, speciation, and stress responses in animals that might be similar to those observed in the hybrid cell line.

Long et al. (2016) [36] also showed the presence of species-specific hypomethylated regions that exist in mice and humans that were CpG-rich, involving young repetitive elements. Based on them, it was due to the absence of a mechanism in driving the methylation to these sites. This finding agreed with the hypomethylated Arabidopsis repetitive elements in the human background in this study (Figure 4).

### 3.4. No Preferential Expression of Original Arabidopsis Genes in Human Genome Background

In this study, six (*NPY4*, *CA2*, *LHCB4.3*, *AtNADH*, *GAMMA-TIP*, and *PLP6)* of the ten *Arabidopsis* genes were expressed in the hybrid cells (Figure 5 and Figure 6), whereas non-expression was not related to gene function. Long et al. (2016) [36] showed that recapitulation of the original hypomethylation states in DNA inserts did not correspond to a change in gene expression in the human and mouse host cells, where even methylation changes in the promoter regions did not alter the gene expression patterns. Previous studies have shown that chloroplast- and photosynthesis-related genes have the most enriched gene ontology across various plant species, highlighting their highly conserved nature and significance in plants [51]. Therefore, we analyzed the expression of photosynthetic and cell wall-related genes in the hybrid cell lines (Table 2 and Figure 6). The photo-responsive *NPY4* gene located on *Arabidopsis* chromosome 2 was found to be expressed only in the hybrid cell lines but not in wild-type *Arabidopsis* leaf tissue (Figure 6). *NPY4*, responsible for the root gravitropic response, is highly expressed in the primary roots and was therefore not expressed in the differentiated leaf tissue [52]. The expression of *NPY4* in the hybrid cells could be attributed to the use of an undifferentiated plant protoplast for cell fusion to generate the hybrid cell line.

Another gene encoding the chloroplast enzyme carbonic anhydrase 2 (CA2) on *Arabidopsis* chromosome 5 was expressed in both the hybrid cells and wild-type *Arabidopsis* plants (Figure 6). Although humans also possess *CA2* with similar functions, the genes evolved independently and belong to different classes—human CA2 belongs to the ⍺-class and plant CA2 to the β-class. The conserved domains for both classes vary; humans use the alpha carbonic anhydrase domain, whereas plants use the beta carbonic anhydrase domain [53]. Therefore, functional homology cannot explain the *CA2* expression in the hybrid cells. The reason for the plant-specific gene expression is unclear, although the ability to express photosynthetic genes in a human–plant hybrid cell line is noteworthy.

## 4. Materials and Methods

### 4.1. Cell Culture

The human–*Arabidopsis* hybrid cell line was obtained from 60- and 149-day-old cultures by Wada et al. (2017). The 60-day-old and 149-day-old cell cultures originated from cell stocks frozen at −80 °C for 5.5 and 4.5 years, respectively. The 300-day-old culture originated from cells grown from the 149-day-old freeze-preserved culture. The hybrid cell line was cultured in Gibco Dulbecco’s modified Eagle’s medium (Thermo Fisher Scientific, Waltham, MA, USA) supplemented with Gibco 10% fetal bovine serum (Thermo Fisher Scientific) and 6-μg/mL blasticidin S (KNF, Tokyo, Japan) at 37 °C in a 5% CO_2_ incubator and subcultured every 2 to 3 days.

### 4.2. Plant Material and DNA Extraction

The seeds of *Arabidopsis thaliana* cv. Columbia (2*n* = 10) were sterilized in 1 mL of 10% (*v/v*) kitchen bleaching solution (KAO, Tokyo, Japan) for 30 min, rinsed several times with distilled water, vernalized at 4 °C for 2 to 3 days, and germinated on 0.5× Murashige and Skoog medium (Fujifilm-Wako, Tokyo, Japan) supplemented with 0.5% agar for 10–15 days at 25 °C. DNA was extracted from seedlings in DNA Suisui buffer (RIZO Inc., Tsukuba, Japan) and precipitated using ethanol.

### 4.3. Whole-Genome Bisulfite Sequencing (WBGS)

DNA extraction from 300-day-old hybrid cell culture was performed as described previously [26] following Miller et al. (1988) [54]. Approximately 8 million cells at 90% confluence in 10-cm plates were lysed using 300-µL nuclei lysis buffer (10-mM Tris-HCl, 400-mM NaCl, and 2-mM EDTA, pH 8.2) and digested overnight at 37 °C with 20-µL 10% sodium dodecyl sulfate (SDS) and 50-µL proteinase K solution (1-mg protease K in 1% SDS and 2-mM EDTA). Thereafter, 100 µL of 5-M NaCl was added and shaken vigorously for 15 s to precipitate the proteins. The sample was pelleted at 13,000× *g* for 15 min, and the supernatant was transferred to a new tube. Next, 10-µL RNase A (10 mg/mL) was added to 200 µL of the solution and incubated at 37 °C for 1 h. Two volumes of absolute ethanol were added, and the tube was inverted several times and incubated overnight at −80 °C. The sample was then centrifuged at 13,000× *g* for 30 min at 4 °C. The supernatant was discarded, and the pellet was washed with 70% cold ethanol and centrifuged again at 13,000× *g* for 5 min at 4 °C. The supernatant was discarded, and the pellet was left to air-dry before being resuspended in distilled water. DNA concentration was measured using a Qubit™ 4 fluorometer (Thermo Fisher Scientific), and its integrity was checked via gel electrophoresis.

WGBS was performed by Macrogen Corp Japan on an Illumina NovaSeq 6000 with 150 Gb of 151-bp paired-end sequencing. The sequencing library was prepared using the Accel-NGS Methyl-Seq DNA library kit (Swift BioSciences, Ann Arbor, MI, USA) and EZ DNA Methylation-Gold kit (Zymo Research, Irvine, CA, USA) according to the library protocol for Illumina platforms. Unmethylated lambda phage DNA (Promega, Madison, WI, USA) was added to the DNA prior to fragmentation as a spike-in control to estimate the bisulfite conversion rates.

### 4.4. WGBS Data Processing

Raw sequence reads were filtered based on quality, and adapter sequences were trimmed using Trim Galore (0.4.5) [55]. Eighteen base pairs were trimmed from the 3′ ends of R1 and 5′ ends of R2 to eliminate the majority of the adaptase tails after adaptor trimming, and reads shorter than 20 bp were discarded. The trimmed reads were mapped to TAIR10 using BSMAP v2.87 [27]. The uniquely mapped reads were sorted, indexed, and divided by PCR duplicates using SAMBAMBA (v0.59) [56]. De-duplicated reads were then passed to methylation calling to assign the methylation states for each base. The alignment was evaluated using Qualimap 2.2 [57]. The cytosine methylation ratio was extracted from the mapping results using the “methratio.py” script from BSMAP with a cut-off value of 1 count. Coverage profiles were calculated from cytosine in the context of CG, CHH, and CHG. Each cytosine location was annotated using the table browser function of the UCSC genome browser (http://genome.ucsc.edu/, accessed on 19 May 2021). A methylation analysis was also performed using Bismark (v0.20.0) [28] and the trimmed reads, as described previously herein. The distribution of the methylation patterns across the gene bodies and the 2-kb upstream and downstream sequences were analyzed and plotted using deeptools [58]. The reference bed files were obtained from the UCSC genome browser Ensembl genes 44 track and LTR track under the ReMap Regulatory Atlas (http://genome.ucsc.edu/, accessed on 19 May 2021).

### 4.5. Differential Methylation Analysis

The publicly available wild-type *Arabidopsis* WGBS sequence data were obtained from SRA (https://www.ncbi.nlm.nih.gov/sra, accessed on 19 May 2021) under accession numbers SRR7586644 and SRR534177. They were quality-checked with FastQC (V0.11.8) [59] and processed using Bismark, similar to the hybrid cell line data, converted to bedgraph format using bismark2bedGraph, and used for the downstream analysis. DMRs between the hybrid cells and wild-type *Arabidopsis* were identified using the methylKit R [31]. The methylation differences were determined using Fisher’s exact test filtered by a minimum difference of 0.25 and q-value of 0.01. A tiling window analysis was performed with a window size of 1000 and a step size of 1000. The methylation calls were filtered by discarding the bases with coverage below 5 and bases that had more than 99.9th percentile coverage in each sample. Only bases that were present in both the control and hybrid cells were retained. The distribution of DMRs among the different chromosomes and gene regions was obtained using methylKit statistics. The DMRs were exported as bedgraphs and analyzed for repeat element enrichment via overlapping with RepeatMasker tracks on the ReMap database [32] using the BEDTools intersect tool [60].

### 4.6. Gene Expression Analysis

RNA was extracted from the hybrid cells using the TRIzol reagent (MRC, Cincinnati, OH, USA) according to the manufacturer’s protocol. For this, 2-µg RNA was used for cDNA synthesis using the ReverTra Ace qPCR RT master mix with gDNA remover (Toyobo, Tokyo, Japan). The cDNA was then processed for PCR quantification. Several primer pairs were designed using http://atrtprimer.kaist.ac.kr, accessed on 19 May 2021 [61] and based on a previous publication by Wada et al. (2017) [25] (as listed in Appendix A). The gene function and homology were obtained from the *Arabidopsis* Information Resources (TAIR) (arabidopsis.org, accessed on 19 May 2021) and UniProt (uniprot.org, accessed on 19 May 2021) databases.

## 5. Conclusions

Our findings demonstrate that some DNA methylation states can be maintained even in a highly divergent background, such as the case of plant DNA recombined with the human genome. We do not have the original *Arabidopsis* protoplast to use as a control for this study. However, as a general control for this study, we believe that a comparison of the recombinant plant chromosomal fragments with wild-type *Arabidopsis* would be a reasonable way to compare the behavior of the plant *Arabidopsis* DNA between a wild type and a hybrid. In conclusion, this study demonstrated the conservation of general CG methylation across evolutionarily distant organisms and the alterations to CG methylation in the plant centromere on the recombinant chromosomal fragments in the hybrid cells. This will aid in exploring the possibility of the conservation of epigenetic control mechanisms across distantly related organisms in the field of molecular biology.

## Figures and Tables

**Figure 1 ijms-22-05426-f001:**
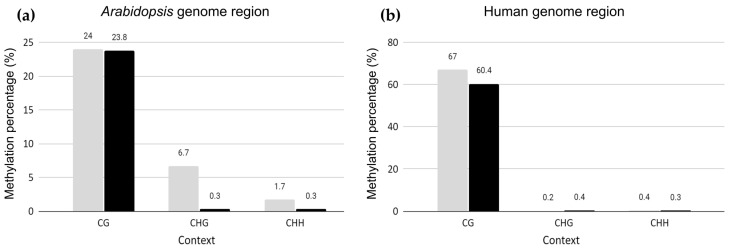
Methylation ratio of the *Arabidopsis* (**a**) and human (**b**) genome regions in a 300-day-old hybrid cell line compared to the methylation state of their wild-type counterpart (*Arabidopsis*: Cokus et al., 2008; HT1080: Wong et al., 2016). Gray: wild-type and black: hybrid cell line.

**Figure 2 ijms-22-05426-f002:**
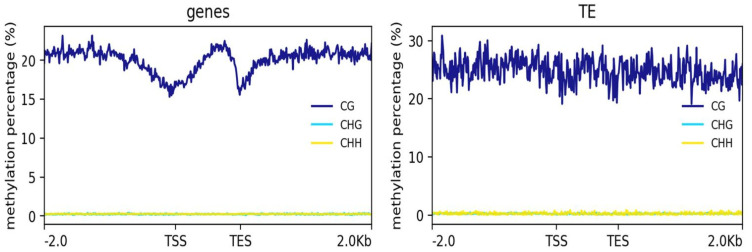
The extent of the methylation of in the context of CG, CHG, and CHH in the introgressed *Arabidopsis* genes and transposable elements (TEs) in the hybrid cells. TSS: transcription start site and TES: transcription end site.

**Figure 3 ijms-22-05426-f003:**
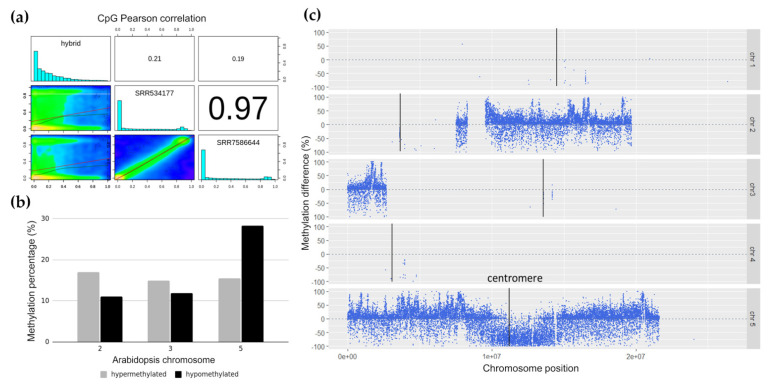
Differential methylation of the *Arabidopsis* genomic regions in the hybrid cells compared to that in the wild-type *Arabidopsis* bisulfite genome data. (**a**) Sample correlation between wild-type *Arabidopsis* (SRR534177 and SRR7586644) and the hybrid cell line. (**b**) Chromosomal distribution of the differentially methylated regions, with hypomethylated (gray) and hypermethylated (black) regions. (**c**) Distribution of the differentially methylated regions on the *Arabidopsis* chromosomes.

**Figure 4 ijms-22-05426-f004:**
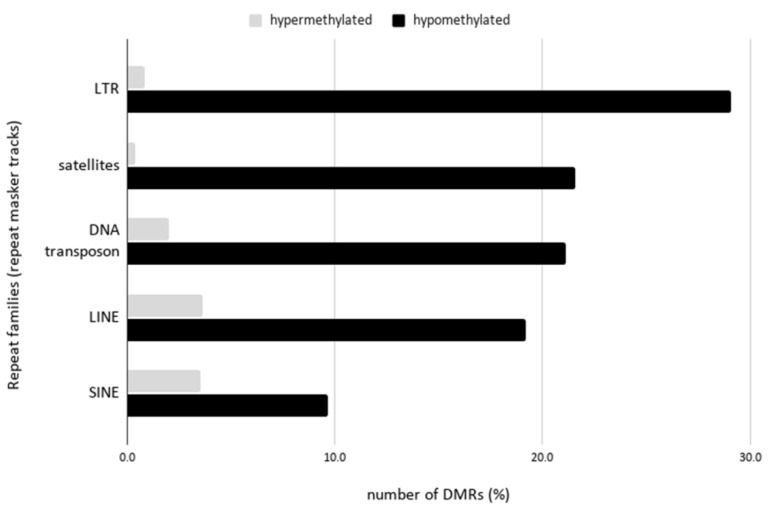
Numbers of differentially methylated *Arabidopsis*-specific repeats in the 300-day-old hybrid cell line compared to that in the wild-type *Arabidopsis* data (SRR7586644). Hypermethylated, gray; hypomethylated, black. LTR: long terminal repeat, LINE: long interspersed nuclear element, and SINE: short interspersed nuclear element.

**Figure 5 ijms-22-05426-f005:**
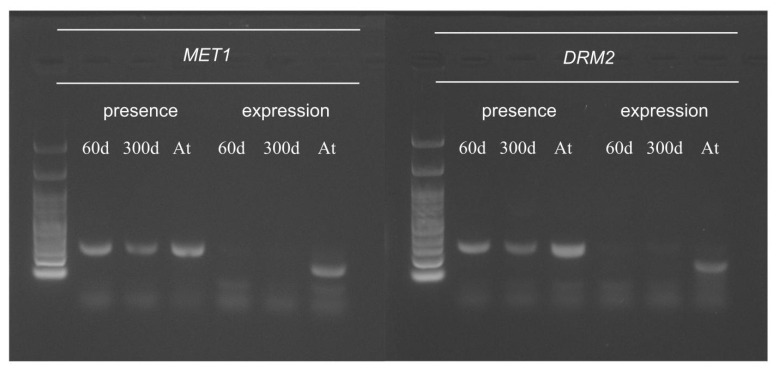
Presence and expression of *Arabidopsis* methyltransferases in the 60- and 300-day-old hybrid cell lines. 60d: 60-day-old hybrid cell line, 300d: 300-day-old hybrid cell line, and At: wild-type *Arabidopsis* plant.

**Figure 6 ijms-22-05426-f006:**
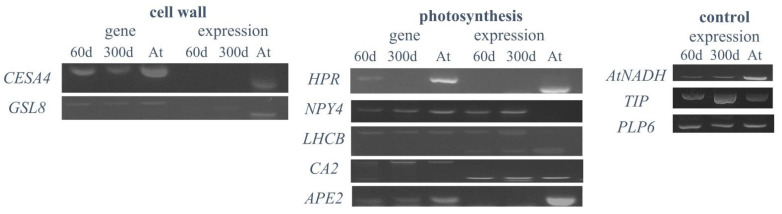
Presence and expression of plant-specific genes in the 60- and 300-day-old hybrid cell lines. 60d: 60-day-old hybrid cell line; 300d: 300-day-old hybrid cell line; At: wild-type *Arabidopsis* leaf plant.

**Table 1 ijms-22-05426-t001:** Presence and expression of *Arabidopsis* maintenance methyltransferases that recognize different cytosine contexts.

*Arabidopsis* Methyltransferase	Methylation Context	Presence	Expression
DDM2/MET1	CG	Yes	No
CMT3	CHG	No	No
DRM2	CHH	Yes	No

**Table 2 ijms-22-05426-t002:** *Arabidopsis* genes used for the expression analysis in the hybrid cell lines.

Gene	Locus	Encoding Protein	Gene Ontology	Log_2_ Ratio *	Expression in 60-Day-Old Cells	Expression in 300-Day-Old Cells
*HPR*	AT1G68010	Hydroxyperuvase reductase	Cellular response to light stimulus, photorespiration, chloroplast	3.22	N	Gene absent
*NPY4*	AT2G23050	Naked Pins in YUC Mutants 4	Positive gravitropism	7.96	Y	Y
*LHCB4.3*	AT2G40100	Light Harvesting Complex Photosystem II	Photosynthesis, light harvesting, response to light stimulus, chloroplast	4.22	Y	Y
*CA2*	AT5G14740	Carbonic anhydrase 2	Carbon utilization, chloroplast	9.06	Y	Y
*APE2*	AT5G46110	Acclimation of Photosynthesis to Environment 2	Photosynthetic acclimation, chloroplast	7.01	N	N
*CESA4*	AT5G44030	Cellulose Synthase A4	Cell wall biogenesis	-	N	N
*GSL8*	AT2G36850	Glucan Synthase-Like 8	Pollen development	-	N	N
*AtNADH*	AT5G11770	NADH-ubiquinone oxidoreductase	Aerobic respiration	11.23	Y	Y
*GAMMA-TIP*	AT2G36830	Gamma-tonoplast intrinsic protein 1	Transmembrane transport, response to salt stress	10.44	Y	Y
*PLP6*	AT2G39220	Patatin-like protein 6	Hydrolase activity	10.28	Y	Y

* Log_2_ ratio data are from Wada et al. (2017) [25]. Y: expressed and N: not expressed. Locus, encoding protein, and gene ontology were from the TAIR database.

## Data Availability

The raw WGBS data that supported the findings of this study are available under the SRA database at https://www.ncbi.nlm.nih.gov/sra, accessed on 19 May 2021, with the biosample accession number SAMN18252892 and bioproject accession number PRJNA713491. The *Arabidopsis* control datasets were obtained from SRA under accession numbers SRR7586644 and SRR534177.

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
