# Peer review of "Epigenetic Distribution of Recombinant Plant Chromosome Fragments in a Human–Arabidopsis Hybrid Cell Line"

_ijms, 2021, doi:10.3390/ijms22115426_

Round 1
Reviewer 1 Report
I have a very mixed feeling for this article. Authors putting Plant DNA in human-Arabidopsis hybrid cell line may pose a challenge being it’s an unrealistic experiment. I am not able to predict its future therapeutic application for humanity. Although authors have interesting findings it shows a lack of scientific rationale of doing the experiment. Authors must put a good argument of why they did this experiment and what is the mileage out of using a heterologous system.
Arabidopsis genome fragments (~30 Mb) were maintained in human cell background; did authors characterize the hybrid cancer cell line? Why did not author's preferred protoplast culture where they could have inserted human nucleus in Arabidopsis protoplast? Was the 30MB genome maintained as epigenome or chromosome insert throughout the cell division, what was doubling time? What are the characteristics of human cell loss by plant genome insertion? Have you done karyotyping for DNA brakes at plant vs human junction? Was DRM2 and MET1 from Arabidopsis also inserted in the human genome.
Can authors show any plant specific proteins by western to make the case of this article? Please use the Human cell line as negative control, Arabidopsis (may be protoplast) cell extract as positive control and hybrid cells as an experimental. I would prefer photosynthesis complexes to do western. Authors are free to choose any plant specific protein which is inserted in hybrid cells. May be figure 6 candidates as western is the best.
I am sure your institutional ethics department should not have any problem with this kind of experiment.
Reviewer 2 Report
Yeng Mun Liaw and colleagues describe in the manuscript at hand how DNA methylation changes if foreign genomes (here ~30 MB of A. thaliana) are inserted into a host cell line (here H. Sapiens). The authors mainly focus on changes observed between the inserted sites and the original locations in the full genome in its physiological context (wild type A thaliana.) Similar experiments have been performed previously involving different species. Accordingly, most of the results presented are not surprising and/or do not yield novel insight. Due to the absence of the plant methylation machinery the plant specific methylation is largely lost. Conversely, the inserted regions adopt in part methylation features of the host system. Some of the more interesting results are the loss of methylation on plant repetitive regions and also some impact on host methylation, i.e. a reduction in CG methylation.
What this study most importantly lacks is replication. An independently generated hybrid line potentially involving different plant genome fragments would allow an assessment of the general robustness of the observed effects. But as this would be too much to ask for the manuscript might be improved by addressing some issues more in deep to increase the impact of this study.
1) In addition to assessment of bulk CG methylation which largely does not change could the authors investigate if there are site- and context-specific differences that could help explain why the arabidopsis fragments do not acquire similar CG methylation levels compared to the host cell? In other words, do the plant regions lack context? Do maybe some few plant sites with human context acquire methylation?
2) As regards the repetitive elements could hypomethylation be explained by a lack of siRNA-production from the repeats in the human context? I would like to see a some more discussion here.
3) I do not appreciate the term landscape in the context of this study as just aggregate analyses have been conducted which do not describe landscapes as such. "Distribution" would be a better term.
Round 2
Reviewer 1 Report
Authors should have more compelling evidence of justifying this hybrid cell line. Authors should also work on karyotyping the chromosome for checking any possible brakes.